# Lung Transplantation in Idiopathic Pulmonary Fibrosis Patients in the European MultiPartner IPF Registry: Challenges for Health Equity

**DOI:** 10.3390/biomedicines13112684

**Published:** 2025-10-31

**Authors:** Nóra M. Tóth, Mordechai R Kramer, Martina Šterclová, Veronika Müller, Katarzyna B. Lewandowska, Nesrin Mogulkoc, Marta Hájková, Michael Studnicka, Jasna Tekavec-Trkanjec, Sanja Dimic-Janjic, Anton Penev, Zoran Arsovski, Jakub Gregor, Petra Ovesná, Martina Koziar Vašáková

**Affiliations:** 1Department of Pulmonology, Semmelweis University, 1083 Budapest, Hungary; 2Institute of Pulmonary Medicine, Rabin Medical Center, Petah Tikva 49100, Israel; 3Department of Respiratory Medicine, Thomayer University Hospital, 14059 Prague, Czech Republic; 4First Department of Pulmonary Diseases, Institute of Tuberculosis and Lung Diseases, 01-138 Warsaw, Poland; 5Department of Pulmonary Medicine, Ege University Medical School, Izmir 35100, Turkey; 6Clinic of Pneumology and Phthisiology, University Hospital Bratislava, 826 06 Bratislava, Slovakia; 7Department of Respiratory Medicine, Paracelsus Medical University, 5020 Salzburg, Austria; 8Pulmonary Department, University Hospital Dubrava, 10 000 Zagreb, Croatia; 9Faculty of Medicine, University of Belgrade, 11000 Belgrade, Serbia; 10Clinic for Pulmonology, University Clinical Center of Serbia, 11000 Belgrade, Serbia; 11Pulmonary Department, Acibadem City Clinic Tokuda Hospital Sofia, 1407 Sofia, Bulgaria; 12PHI University Clinic of Pulmonology and Allergy, Faculty of Medicine, University Ss Cyril and Methodius, 1000 Skopje, North Macedonia; 13Institute of Biostatistics and Analyses, Faculty of Medicine, Masaryk University, 62500 Brno, Czech Republic

**Keywords:** idiopathic pulmonary fibrosis, lung transplantation, healthcare access, geographic disparities, transplant referral

## Abstract

**Background:** Despite advancements in pharmacological therapy, lung transplantation (LuTX) remains the only life-prolonging treatment in end-stage idiopathic pulmonary fibrosis (IPF). However, real-world referral patterns in Central and Eastern European (CEE) countries remain poorly characterized. We aimed to comprehensively review factors influencing referral and identify systemic barriers to LuTX access. **Methods:** Baseline characteristics of IPF patients potentially eligible for LuTX, enrolled in the European MultiPartner IPF Registry between 2012 and 2022 (n = 1256), were retrospectively analyzed. LuTX (n = 94) and potentially eligible but not transplanted (n = 1162) subgroups were compared. National experts also completed a questionnaire assessing transplant referral and listing practices across different healthcare systems. **Results:** Only 7.5% of potentially eligible subjects were transplanted, revealing substantial geographic disparities, with Israel having the highest rates (43.1%), followed by Austria (9.5%), Hungary (7.8%), and the Czech Republic (4.6%). LuTX patients were younger (60.2 ± 7.4 vs. 62.6 ± 6.2 years, *p* < 0.001), had worse lung function (FVC 60 ± 15 vs. 74 ± 21% predicted; *p* < 0.001, TLCO 41 ± 15 vs. 49 ± 19% predicted; *p* < 0.001), and were more likely to receive antifibrotic and oxygen therapies. The most frequent reasons for exclusion from referral/listing were age > 70 years and concomitant heart/renal failure. **Conclusions:** This first comprehensive CEE analysis demonstrates low IPF transplant rates with high inter-country variability. Patients presenting early with functionally advanced disease are more likely transplanted, while advanced age remains the primary exclusion factor, highlighting critical access gaps potentially contributing to regional outcome differences.

## 1. Introduction

Idiopathic pulmonary fibrosis (IPF) is a progressive and fatal interstitial lung disease characterized by the relentless scarring of lung tissue, leading to impaired respiratory function and compromised quality of life [1,2]. Despite that antifibrotic therapy has shown to slow down lung function deterioration in IPF, lung transplantation (LuTX) remains the only treatment option for eligible end-stage patients that increases survival and long-term quality of life [3]. Therefore, the current guideline of the International Society of Heart and Lung Transplantation (ISHLT) includes referral for transplantation at the time of diagnosis for possible eligible patients [4]. However, the selection criteria for LuTX are multifaceted and both advanced age and comorbidities are important relative and/or absolute contraindications [4]. LuTX programs have been started during the last few decades in the Central and Eastern European (CEE) region and access to it is not equal for IPF patients.

The European MultiPartner IPF REgistry (EMPIRE) represents an invaluable resource for exploring possible factors influencing the decision-making on whether an IPF patient might undergo LuTX in the CEE region. This comprehensive, international registry gathers detailed real-world patient information from a diverse cohort of individuals diagnosed with IPF from 11 countries, capturing essential clinical, radiological, and functional data [5,6].

In the current study, we aimed to identify the differences in baseline patient characteristics between IPF patients who received LuTX and those who did not, according to the EMPIRE data. We also explored the disparities in patient selection practices for LuTX among the participating countries. Our findings could assist in refining patient selection criteria and help decision-making for healthcare professionals and IPF patients at different stages of disease progression in this region. Identifying inter-country differences might also facilitate changes in the current protocols of healthcare providers to provide more equal care for this population.

## 2. Materials and Methods

### 2.1. Study Design

In our retrospective cohort analysis, LuTX referral and listing practices of the EMPIRE countries were evaluated by applying the ISHLT selection criteria for LuTX candidates [4] (adapted to the format and availability of data in the registry) to the baseline data from all patients enrolled in the registry. Furthermore, country-specific aspects for referral/listing were also collected from countries with transplanted patients in the EMPIRE using a short questionnaire.

### 2.2. Study Population and Collected Data

All validated IPF patients enrolled in the EMPIRE between March 2012 and May 2022 were included [1,7]. Overall, 4390 cases were included from 11 countries from the EMPIRE, namely Austria, Bulgaria, Croatia, the Czech Republic, Hungary, Israel, North Macedonia, Poland, Serbia, Slovakia, and Turkey (see all participating centers by country in the Appendix A). The following data were collected: demographics (age, height, weight, sex), date of diagnosis, clinical signs and symptoms, radiological and histopathological patterns, the Gender–Age–Physiology (GAP) index, which is used for predicting clinical severity in IPF patients [8], serum autoantibody profile, smoking history, history of respiratory infections (sporadic: less than once a year or frequent: more than once a year), IPF treatment (pharmacological and non-pharmacological, such as pulmonary rehabilitation, long-term oxygen therapy, and lung transplantation), comorbidities and non-IPF medications, and longitudinal follow-up data on lung function and carbon monoxide diffusion capacity of the lung, as well as whether a patient was considered for LuTX. Data were individually uploaded by each center of the registry and reported anonymously [6]. Although data entry was performed locally, all centers followed internationally accepted guidelines and standardized procedures for the diagnosis and management of IPF.

### 2.3. Subgroups of Eligible Patients with and Without LuTX

We assessed and compared the baseline characteristics of patients with successful LuTX (n = 94) and those patients who potentially could have been eligible for LuTX but were not transplanted (n = 1162). As the entire EMPIRE cohort is different from the cohort of potentially transplantable patients, exclusion criteria were determined based on the ISHLT guideline [4] (Table 1). The detailed patient selection process is summarized in Figure 1. The baseline is defined as the date of entry into the EMPIRE.

### 2.4. Assessment of LuTX Referral and Listing Practices Using a Questionnaire

In order to assess the differences in the LuTX referral and listing practices, a short questionnaire was used (Appendix A). Responses representing the general opinion of the local LuTX teams were collected (see the list of responding centers by country in Appendix A).

In the first half of the questionnaire, local experts had to make country-specific rankings among the factors that influence their decision to not refer an IPF patient for LuTX (responses from 5 countries could be evaluated, including the Czech Republic, Hungary, Israel, Poland, and Serbia). The following 14 factors were ranked regarding referral: age > 65 years, age > 70 years, current smoker, social deprivation, comorbidity of heart failure, comorbidity of renal failure, comorbidity of osteoporosis, comorbidity of liver cirrhosis, cancer any type, positron emission tomography-positive changes requiring additional invasive investigations, positive panel-reactive antibodies, autoimmune serology positivity, body mass index (BMI) > 35 kg/m^2^, or BMI < 16 kg/m^2^.

Second, participating centers could also freely specify the 5 most important aspects that led to the exclusion of IPF patients from listing for LuTX (responses from 4 countries could be evaluated, namely Hungary, Israel, Poland, and Serbia).

### 2.5. Ethical Statement

This study was conducted in accordance with the Declaration of Helsinki. The EMPIRE protocol was approved by the respective ethical committees of the individual countries and sites involved in the registry. All patients signed an informed consent form prior to enrollment in the registry.

### 2.6. Statistical Analysis

R software 4.1.1. was used for statistical analysis. Data were presented as median (interquartile range) for continuous variables and absolute and relative frequencies for categorical variables. Significant differences among the groups were analyzed by Mann–Whitney or Fisher’s exact tests. *p* < 0.05 was considered statistically significant.

## 3. Results

### 3.1. Lung Transplantation Rate and Exclusion Factors

A total of 94 patients from the EMPIRE underwent LuTX. In total, 3134 (73%) of the non-transplanted patients met at least one exclusion criterion and thus were excluded from the study. The most common reason was older age (50.2%), followed by the patient’s refusal of LuTX (15.2%) and malignancy (13.2%) (Table 1). Of note, some exclusion criteria (refused LuTX, age > 70 years, being extremely obese/underweight, current smoker, presence of malignancy, concurrent antiplatelet, and anticoagulant therapy) were also present in the transplanted group, and they still underwent surgery. This can be explained partly because they were assessed at inclusion to the registry and were modifiable factors (smoking, BMI, transplant refusal, becoming cancer-free) or because they were relative contraindications (age, antiplatelet and anticoagulant therapy).

According to EMPIRE entries (Table 2), 30.9% of patients who underwent transplantation were initially not considered candidates for LuTX.

Figure 2 shows the proportion of transplanted and not transplanted patients in the subpopulation potentially eligible for LuTX. Overall, 7.5% of the eligible subjects have been transplanted. Among the EMPIRE countries, Israel had the highest transplantation rate (43.1%), followed quite far behind by Austria (9.5%) and Hungary (7.8%).

### 3.2. Comparison of Clinical Characteristics Between Eligible Patients with and Without LuTX

Table 3 demonstrates the distribution of patients with and without LuTX among the countries and baseline characteristics of the transplanted and the potentially eligible but not transplanted cohorts. Transplanted patients were younger both at the time of diagnosis and at the time of enrollment to the registry and experienced symptoms longer than those who did not undergo LuTX. They also had more comorbidities than the non-transplanted group; in particular, gastrointestinal diseases and hematological and immune diseases were more frequent in transplanted patients.

Transplanted patients presented with worse baseline lung function and CO diffusion capacity, as well as a higher GAP index. Also, they were more often given antifibrotics at the time of inclusion and were more likely to require long-term oxygen therapy and participate in pulmonary rehabilitation programs than non-transplanted patients. Finger clubbing, exertional dyspnea, and cough were more common in the transplanted population.

Signs of connective tissue diseases were more frequent in transplanted patients; however, the presence of autoantibodies did not differ between the two cohorts. The transplanted group received systemic corticosteroid treatment more often. Respiratory infections and inorganic dust exposure were more prevalent in transplanted subjects as well.

### 3.3. Patient Characteristics by Country

To identify possible factors causing the striking differences in the number of transplanted patients among the EMPIRE countries, we collected the most important characteristics by country in Table 4. Although the uneven distribution of patients with LuTX among the countries did not allow us to perform robust inter-country statistical comparisons, it can be seen that patients in Turkey and Israel had worse, while Poland showed noticeably better lung function parameters; also, in Poland and in Hungary, diffusion parameters were better than in the other EMPIRE countries.

### 3.4. Clinical Factors Influencing Referral for LuTX by Experts

Among the pre-set factors which could potentially influence referral, experts named heart failure as the most important determinant for not referring an IPF patient for LuTX evaluation in the EMPIRE countries, followed by renal failure and liver cirrhosis. Age > 70 years and malignancy proved to be the fourth most important determinant (Figure 3). When experts were allowed to define the most influential factors for not listing a referred patient, the most frequently named reasons were heart failure and age > 70 years, followed by renal failure and malignancy in the last 3–5 years.

## 4. Discussion

IPF has become one of the leading causes for LuTX worldwide [9], yet data on LuTX of IPF patients in the CEE region were not available. We found that despite a substantial proportion of EMPIRE patients proving to be potentially eligible for LuTX, only a small fraction (7.5%) underwent transplantation, with considerable inter-country differences.

Our results align with data from the United States IPF-PRO registry, which reported that only 96 out of 955 IPF patients (10.1%) received LuTX, while just 25.9% were ever evaluated for LuTX [10]. An earlier analysis from the same registry showed that 23.3% of the eligible patients were referred to LuTX [11]. Similarly, another United States registry found that only 8.8% of IPF patients were listed for or underwent LuTX. [12] However, the several-fold difference observed between countries in our cohort (43.1% vs. 0.0–9.5%) represents an unprecedented level of geographic disparity, far exceeding previously documented variations. These findings highlight the need to improve referral and listing practices in the region, especially considering the extraordinarily high rates (43.1% transplanted) Israel has been able to achieve, while much lower numbers were observed in other countries. According to current international guidelines, the referral of IPF patients should be made at the time of diagnosis, even if pharmacological therapy is initiated, and detailed evaluation is recommended at the first sign of objective deterioration [2,4].

Younger age and worse baseline lung function were present in the transplanted subjects compared to the non-transplanted group, which suggests that patients with early presentation of a more advanced disease are more likely to receive LuTX. US data have also described younger age, decreased FVC, and oxygen use as predictors of LuTX referral [11] and listing [12]. In line with previous data, advanced age and comorbidities were shown as the main reasons for not referring patients in the EMPIRE countries. Although age higher than 70 years is considered as a factor with high risk for poor LuTX outcome, there is no age limit as an absolute contraindication according to current international guidelines [4]. And while older patients are at greater risk of frailty and potentially contraindicating comorbidities [13], data show that post-transplant survival of patients over 70 has improved significantly in the past years, and carefully selected elderly subjects may have similar short-term post-transplant outcomes than younger recipients [14]. Life expectancy in CEE countries is variable and generally lower than in more developed countries [15]. As the post-transplant median survival in IPF is estimated to be around 4.5 years [16], most of these countries use lower age limits as a relative contraindication for transplant. However, this conservative age approach in CEE countries contradicts emerging evidence demonstrating meaningful survival benefit for single LuTX in IPF patients over 65 years of age, with no difference in survival following transplant, rates of primary graft dysfunction, or acute/chronic rejection when compared to younger recipients [17]. Our finding that age > 70 years remains a leading exclusion criterion suggests that CEE countries may be missing opportunities to extend survival for a significant patient population.

Although LuTX was more often performed in patients diagnosed with IPF at an earlier age, comorbidities were abundant in these patients, possibly due to more comprehensive data available from pre-LuTX assessments. Considering that a substantial proportion (30.9%) of the transplanted patients were initially not considered for LuTX (see Table 2), partly due to comorbidities and partly due to other reasons yet were eventually transplanted, there could be a subjective overemphasis on comorbidities by the treating physicians. It would be important for referring physicians to more systematically track patients’ inclusion and exclusion criteria and to re-evaluate them over the course of the disease, particularly in light of the adequate management of comorbid conditions.

Our findings reveal healthcare system-level disparities that extend beyond individual patient factors. Research confirms substantial equity gaps in access to LuTX [10,11,12], but our study demonstrates that entire national healthcare systems can create similar barriers. The several-fold difference between Israel and other CEE countries cannot be explained by patient demographics and registry reporting alone, suggesting systemic healthcare delivery differences that call for closer examination and appropriate policy response.

The optimal choice between single or double LuTX for patients with IPF is still a subject of ongoing debate [18,19,20,21,22,23]. Although double LuTX is preferred in most CEE countries, accumulating evidence suggests no significant difference in overall survival between the two techniques [18,19,21], particularly among older recipients [17,20,24]. In Israel, single LuTX is more commonly performed in IPF than double LuTX [23,25]. This could explain their notable success in achieving high LuTX rates and may serve as a direction to follow in the CEE region in order to broaden the scope of LuTX candidates in IPF patients, even in the age group over 65 years.

More frequent respiratory infections in the transplanted group reinforce previous data indicating that a higher bacterial load in bronchoalveolar lavage or lung tissue samples of IPF patients predicts higher risk of disease progression, acute exacerbation, or death [26,27], since a more rapid deterioration can facilitate the listing of the patient. On the other hand, IPF acute exacerbations can be triggered by infections and are often lethal events [2], making lung transplant evaluations difficult in these cases.

The evaluation of autoimmunity is part of the diagnostic process of IPF for the exclusion of a possible underlying connective tissue disease, yet the relevance of autoantibody positivity and in IPF patients is still under debate [28,29]. We did not find any difference in autoantibody serology outcomes between groups; however, patients presenting autoimmune symptoms were more often transplanted, bringing forth a possible association with a more progressive disease course, as opposed to previous findings [30].

Our study has the inherent limitations of a retrospective review of a large, multicentrically collected database. Specific limitations are data entry gaps and potential reporting bias regarding performed LuTXs. Another limitation was that LuTX was a reason to stop follow-ups of patients in the registry, i.e., only data up to the date of LuTX are available for the transplanted cohort. The data structure of the registry also resulted in limitations in terms of possible exclusion criteria for LuTX eligibility. The questionnaire analysis provides only a limited representation of the different healthcare systems within the EMPIRE due to a lack of responses from a substantial number of the participating countries.

## 5. Conclusions

In summary, our data confirm that LuTX can be a treatment option in several patients with end-stage IPF in the CEE region. While reporting in the EMPIRE is not fully representing a given country’s IPF care, these data confirm that more emphasis should be placed on LuTX referral in most countries, as significant regional differences are noted. The experience of experts from Israel, which had the highest rate of eligible patients with LuTX, should pave the way for improved protocols for better care of end-stage IPF. Given that recent evidence supports meaningful survival benefit for LuTX in IPF patients over 65 years of age and the better treatment possibilities of IPF, these findings should raise awareness for more effective healthcare policies to address systematic barriers preventing life-saving transplantation for the majority of eligible IPF patients in the CEE region.

## Figures and Tables

**Figure 1 biomedicines-13-02684-f001:**
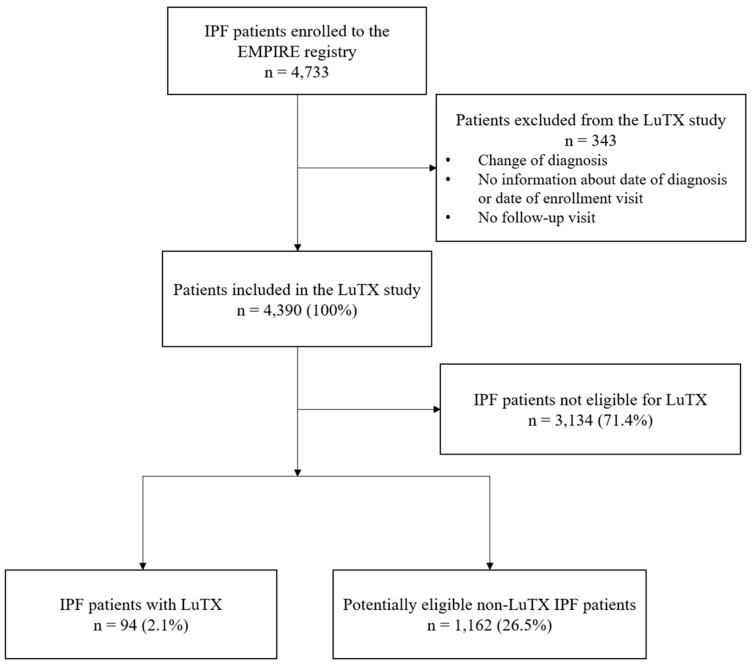
Patient selection. EMPIRE, European MultiPartner IPF Registry; IPF, idiopathic pulmonary fibrosis; LuTX, lung transplantation.

**Figure 2 biomedicines-13-02684-f002:**
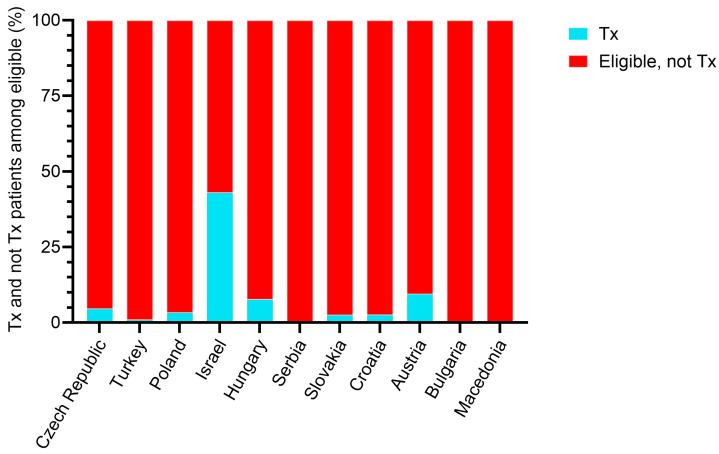
Proportion of transplanted and non-transplanted patients in the IPF population potentially eligible for LuTX based on exclusion criteria among the EMPIRE countries.

**Figure 3 biomedicines-13-02684-f003:**
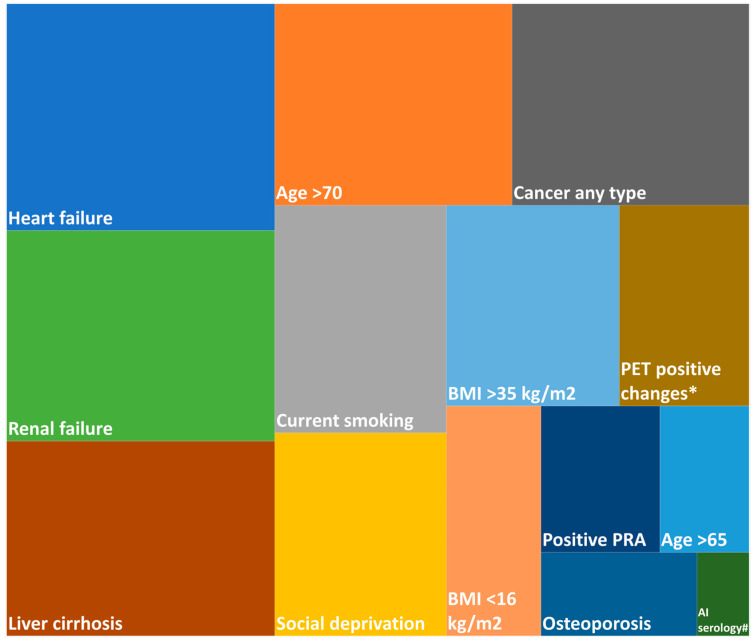
The importance of different factors influencing referral based on the ranking of the questionnaire. The area on the diagram is proportional to the average rank of each factor. * Positron emission tomography-positive changes requiring additional investigations; # autoimmune serology positivity; PRAs, panel-reactive antibodies.

**Table 1 biomedicines-13-02684-t001:** Exclusion criteria for eligibility for LuTX at baseline.

Criteria	Total n = 4390	LuTX n = 94	No TX n = 4296
Refused TX	656 (14.9%)	1 (1.1%)	655 (15.2%)

Age > 70 years	2163 (49.3%)	7 (7.4%)	2156 (50.2%)

Active tuberculosis	93 (2.1%)	0 (0.0%)	93 (2.2%)

Liver cirrhosis	14 (0.3%)	0 (0.0%)	14 (0.3%)

BMI > 35 kg/m^2^ OR < 16 kg/m^2^	348 (7.9%)	8 (8.5%)	340 (7.9%)

Current smoking	213 (4.9%)	5 (5.3%)	208 (4.8%)

Malignancy	575 (13.1%)	8 (8.5%)	567 (13.2%)

Atrial fibrillation	234 (5.3%)	0 (0.0%)	234 (5.4%)

HIV infection	0 (0.0%)	0 (0.0%)	0 (0.0%)

Concurrent antiplatelet AND anticoagulant therapy	205 (4.7%)	5 (5.3%)	200 (4.7%)

Any exclusion criterion	3163 (72.1%)	29 (30.9%)	3134 (73.0%)
Potentially eligible for LuTX according to criteria OR transplanted	1256 (28.6%)	94 (100%)	1162 (27.0%)

Data are presented as n (%). BMI, body mass index; HIV, human immunodeficiency virus; LuTX, lung transplantation.

**Table 2 biomedicines-13-02684-t002:** Patients considered for LuTX at the inclusion to the registry according to EMPIRE entries.

	All EMPIRE Patients n = 4390	All Patients Potentially Eligible for LuTX n = 1256	Patients with LuTX n = 94	Patients Without LuTX n = 1162
**Considered for LuTX initially according to registry entry**				
Yes	1227 (27.9%)	1227 (97.7%)	65 (69.1%)	1162 (100.0%)
No	3163 (72.1%)	29 (2.3%)	29 (30.9%)	0 (0.0%)
**Reason if not considered**				
Age	2163 (68.4%)	7 (24.1%)	7 (24.1%)	-
Comorbidities	347 (11.0%)	7 (24.1%)	7 (24.1%)	-
Refused	370 (11.7%)	1 (3.4%)	1 (3.4%)	-
Other	283 (8.9%)	14 (48.3%)	14 (48.3%)	-

Data are presented as n (%). EMPIRE, European MultiPartner IPF Registry; LuTX, lung transplantation.

**Table 3 biomedicines-13-02684-t003:** Baseline patient characteristics—selected cohorts from the EMPIRE.

Characteristics	All Patients Potentially Eligible for LuTX n = 1256	Patients with LuTX n = 94	Patients Without LuTX n = 1162	*p*-Value *
**Number of patients**				
All countries	1256 (100%)	94 (100%)	1162 (100%)	
Czech Republic	431 (34.3%)	20 (21.3%)	411 (35.4%)	
Turkey	192 (15.3%)	2 (2.1%)	190 (16.4%)	
Poland	177 (14.1%)	6 (6.4%)	171 (14.7%)	
Israel	123 (9.8%)	53 (56.4%)	70 (6.0%)	
Hungary	103 (8.2%)	8 (8.5%)	95 (8.2%)	
Serbia	79 (6.3%)	0 (0.0%)	79 (6.8%)	
Slovakia	77 (6.1%)	2 (2.1%)	75 (6.5%)	
Croatia	38 (3.0%)	1 (1.1%)	37 (3.2%)	
Austria	21 (1.7%)	2 (2.1%)	19 (1.6%)	
Bulgaria	10 (0.8%)	0 (0.0%)	10 (0.9%)	
Macedonia	5 (0.4%)	0 (0.0%)	5 (0.4%)	
**Men**	886 (70.5%)	72 (76.6%)	814 (70.1%)	0.181
**Age at inclusion (years) (n = 1255)**	63.9 (59.4–67.3)	60.7 (55.9–65.6)	64.1 (59.7–67.3)	**<0.001**
**Age at diagnosis (years) (n = 1255)**	63 (58–67)	59 (55–64)	63 (59–67)	**<0.001**
**Duration of symptoms (months) (n = 1212)**	12 (6–24)	24 (9–48)	12 (6–24)	**<0.001**
**Length of follow-up from enrollment (months)**	19 (7–39)	19 (11–36)	19 (6–39)	0.495
**Time from diagnosis to LuTX/last follow-up (months)**	29 (14–53)	33 (20–51)	29 (14–53)	0.090
**BMI (kg/m** ** ^2^ ** **) (n = 1242)**	27.9 (25.4–30.5)	29.4 (26.5–30.9)	27.8 (25.3–30.5)	**0.005**
**Smoking status**				**<0.001**
Never-smokers	487 (39.0%)	31 (33.3%)	456 (39.4%)	
Ex-smokers	758 (60.6%)	57 (61.3%)	701 (60.6%)	
Current smokers	5 (0.4%)	5 (5.4%)	0 (0.0%)	
**High-resolution CT pattern**				0.086
UIP	825 (65.7%)	71 (75.5%)	754 (64.9%)	
Probable/possible UIP	330 (26.3%)	21 (22.3%)	309 (26.6%)	
Inconsistent with UIP/alternative diagnosis	67 (5.3%)	1 (1.1%)	66 (5.7%)	
Unknown	34 (2.7%)	1 (1.1%)	33 (2.8%)	
**Number of comorbidities**	3.00 (1.00–5.00)	4.00 (3.00–6.00)	3.00 (1.00–5.00)	**<0.001**
**Heart and vascular**	830 (66.1%)	68 (72.3%)	762 (65.6%)	0.183
**Pulmonary**	451 (35.9%)	34 (36.2%)	417 (35.9%)	0.956
**Gastrointestinal**	701 (55.8%)	73 (77.7%)	628 (54.0%)	**<0.001**
**Urogenital**	133 (10.6%)	13 (13.8%)	120 (10.3%)	0.288
**Hematologic and immune system**	79 (6.3%)	13 (13.8%)	66 (5.7%)	**0.002**
**Arterial hypertension**	576 (45.9%)	42 (44.7%)	534 (46.0%)	0.811
**Diabetes mellitus**	230 (18.3%)	26 (27.7%)	204 (17.6%)	**0.015**
**Ischemic heart disease**	212 (16.9%)	26 (27.7%)	186 (16.0%)	**0.004**
**Hyperlipidemia**	300 (23.9%)	41 (43.6%)	259 (22.3%)	**<0.001**
**Duodenal ulcer disease**	278 (22.1%)	39 (41.5%)	239 (20.6%)	**<0.001**
**At least one comedication**	991 (78.9%)	73 (77.7%)	918 (79.0%)	0.759
**Number of comedications**	3.00 (1.00–4.00)	3.00 (2.00–5.00)	2.00 (1.00–4.00)	**0.003**
Beta-blockers	285 (22.7%)	20 (21.3%)	265 (22.8%)	0.734
Angiotensin-converting enzyme inhibitor	280 (22.3%)	17 (18.1%)	263 (22.6%)	0.308
Aspirin	251 (20.0%)	26 (27.7%)	225 (19.4%)	0.053
Statins	285 (22.7%)	26 (27.7%)	259 (22.3%)	0.232
Diuretics	193 (15.4%)	12 (12.8%)	181 (15.6%)	0.467
Insulin	209 (16.6%)	21 (22.3%)	188 (16.2%)	0.123
**Pharmacological IPF treatment**	1046 (83.3%)	90 (95.7%)	956 (82.3%)	**<0.001**
Antifibrotics	861 (68.6%)	83 (88.3%)	778 (67.0%)	**<0.001**
Rehabilitation	319 (25.4%)	42 (44.7%)	277 (23.8%)	**<0.001**
Long-term oxygen therapy	465 (37.0%)	77 (81.9%)	388 (33.4%)	**<0.001**
N-acetylcysteine	221 (17.6%)	17 (18.1%)	204 (17.6%)	0.897
Proton pump inhibitors	222 (17.7%)	14 (14.9%)	208 (17.9%)	0.462
Systemic corticosteroids	276 (22.0%)	33 (35.1%)	243 (20.9%)	**0.001**
Azathioprine	91 (7.2%)	9 (9.6%)	82 (7.1%)	0.365
Other cytostatic	20 (1.6%)	2 (2.1%)	18 (1.5%)	0.657
**First antifibrotic drug**				**<0.001**
Pirfenidone	451 (35.9%)	47 (50.0%)	404 (34.8%)	
Nintedanib	410 (32.6%)	36 (38.3%)	374 (32.2%)	
None	395 (31.4%)	11 (11.7%)	384 (33.0%)	
**First antifibrotic therapy duration (months) (n = 513)**	15 (7–29)	23 (11–37)	14 (6–28)	**<0.001**
**FVC (L) (n = 1106)**	2.58 (1.96–3.25)	2.20 (1.75–2.82)	2.62 (1.98–3.28)	**<0.001**
**FVC (%ref) (n = 1090)**	72 (58–87)	60 (48–69)	74 (60–87)	**<0.001**
**TLCO (mmol/kPa/min) (n = 1021)**	3.92 (2.92–5.09)	3.45 (2.37–4.50)	3.98 (2.92–5.14)	**0.015**
**TLCO (%ref) (n = 949)**	47 (35–59)	41 (32–51)	47 (35–60)	**<0.001**
**6 min walk test (m) (n = 546)**	416 (344–480)	392 (334–469)	420 (345–480)	0.356
**GAP index**				**0.014**
GAP I	546 (54.2%)	33 (39.3%)	513 (55.6%)	
GAP II	382 (37.9%)	41 (48.8%)	341 (36.9%)	
GAP III	79 (7.8%)	10 (11.9%)	69 (7.5%)	
**Dyspnea (New York Heart Association grade)**				0.140
I	54 (4.7%)	1 (1.1%)	53 (5.0%)	
II	714 (61.7%)	53 (57.6%)	661 (62.1%)	
III	368 (31.8%)	37 (40.2%)	331 (31.1%)	
IV	21 (1.8%)	1 (1.1%)	20 (1.9%)	
**Velcro-like crackles**	1166 (93.3%)	90 (96.8%)	1076 (93.0%)	0.162
**Finger clubbing**	552 (44.2%)	62 (66.7%)	490 (42.4%)	**<0.001**
**Exertional dyspnea**	1157 (92.2%)	92 (97.9%)	1065 (91.7%)	**0.033**
**Cough**	848 (67.5%)	73 (77.7%)	775 (66.7%)	**0.029**
**Signs of connective tissue disease**	116 (9.2%)	14 (14.9%)	102 (8.8%)	**0.049**
**Autoantibodies**	97 (7.7%)	8 (8.5%)	89 (7.7%)	0.766
**Exposure to organic dusts**	169 (13.5%)	12 (12.8%)	157 (13.5%)	0.839
**Exposure to inorganic dusts**	221 (17.6%)	24 (25.5%)	197 (17.0%)	**0.036**
**Respiratory infections**	218 (17.4%)	25 (26.9%)	193 (16.7%)	**0.012**
**Type of infection**				0.217
Sporadic	149 (69.3%)	20 (80.0%)	129 (67.9%)	
Frequent	66 (30.7%)	5 (20.0%)	61 (32.1%)	

Data are presented as n (%) or median (interquartile range). * Subgroups of with and without LuTX were analyzed with Fisher’s exact and Mann–Whitney tests. In the case of missing data, the number of patients with available data are shown. BMI, body mass index; FVC, forced vital capacity; GAP, Gender–Age–Physiology index; IPF, idiopathic pulmonary fibrosis; LuTX, lung transplantation; TLCO, transfer factor of the lung for carbon monoxide; UIP, usual interstitial pneumonia.

**Table 4 biomedicines-13-02684-t004:** Baseline characteristics of all patients potentially eligible for LuTX by country.

	All Patients Potentially Eligible for LuTX n = 1256	Czech Republic n = 431	Turkey n = 192	Poland n = 177	Israel n = 123	Hungary n = 103	Serbia n = 79	Slovakia n = 77	Croatia n = 38	Austria n = 21	Bulgaria n = 10	Macedonia n = 5
**Age at inclusion (years)**	63.9 (59.4–67.3)	64.4 (60.0–67.4)	63.1 (58.6–66.8)	64.5 (61.6–67.4)	64.4 (59.3–67.6)	62.7 (59.1–67.2)	61.1 (57.1–66.2)	62.3 (55.9–67.2)	63.6 (60.1–67.6)	66.2 (63.1–68.2)	67.5 (64.4–68.5)	55.0 (53.9–65.3)
**Men**	886 (70.5%)	318 (73.8%)	155 (80.7%)	131 (74.0%)	82 (66.7%)	63 (61.2%)	43 (54.4%)	46 (59.7%)	26 (68.4%)	12 (57.1%)	7 (70.0%)	3 (60.0%)
**Antifibrotic treatment**												
Yes	861 (68.6%)	315 (73.1%)	108 (56.3%)	112 (63.3%)	99 (80.5%)	84 (81.6%)	49 (62.0%)	41 (53.2%)	26 (68.4%)	17 (81.0%)	8 (80.0%)	2 (40.0%)
No	395 (31.4%)	116 (26.9%)	84 (43.8%)	65 (36.7%)	24 (19.5%)	19 (18.4%)	30 (38.0%)	36 (46.8%)	12 (31.6%)	4 (19.0%)	2 (20.0%)	3 (60.0%)
**Number of comorbidities**	3.00 (1.00–5.00)	3.00 (2.00–5.00)	3.00 (2.00–5.00)	2.00 (1.00–4.00)	5.00 (3.00–7.00)	2.00 (1.00–3.00)	1.00 (1.00–2.00)	1.00 (1.00–3.00)	4.00 (3.00–5.00)	2.00 (1.00–3.00)	1.50 (1.00–5.00)	0
**Heart and vascular**	830 (66.1%)	297 (68.9%)	122 (63.5%)	114 (64.4%)	80 (65.0%)	64 (62.1%)	58 (73.4%)	46 (59.7%)	27 (71.1%)	14 (66.7%)	8 (80.0%)	0
**Pulmonary**	451 (35.9%)	168 (39.0%)	150 (78.1%)	35 (19.8%)	32 (26.0%)	22 (21.4%)	10 (12.7%)	11 (14.3%)	19 (50.0%)	3 (14.3%)	1 (10.0%)	0
**Gastrointestinal**	701 (55.8%)	268 (62.2%)	91 (47.4%)	101 (57.1%)	104 (84.6%)	41 (39.8%)	20 (25.3%)	34 (44.2%)	29 (76.3%)	10 (47.6%)	3 (30.0%)	0
**Blood and immune system**	79 (6.3%)	27 (6.3%)	14 (7.3%)	7 (4.0%)	15 (12.2%)	6 (5.8%)	4 (5.1%)	5 (6.5%)	0 (0.0%)	1 (4.8%)	0 (0.0%)	0
**Arterial hypertension**	576 (45.9%)	212 (49.2%)	67 (34.9%)	87 (49.2%)	50 (40.7%)	46 (44.7%)	38 (48.1%)	38 (49.4%)	19 (50.0%)	13 (61.9%)	6 (60.0%)	0
**Diabetes mellitus**	230 (18.3%)	80 (18.6%)	37 (19.3%)	35 (19.8%)	34 (27.6%)	13 (12.6%)	10 (12.7%)	11 (14.3%)	7 (18.4%)	1 (4.8%)	2 (20.0%)	0
**Ischemic heart disease**	212 (16.9%)	76 (17.6%)	32 (16.7%)	31 (17.5%)	34 (27.6%)	12 (11.7%)	3 (3.8%)	9 (11.7%)	10 (26.3%)	1 (4.8%)	4 (40.0%)	0
**Hyperlipidemia**	300 (23.9%)	133 (30.9%)	9 (4.7%)	48 (27.1%)	80 (65.0%)	5 (4.9%)	0 (0.0%)	11 (14.3%)	6 (15.8%)	8 (38.1%)	0 (0.0%)	0
**FVC (L)**	2.58 (1.96–3.25)	2.74 (2.14–3.37)	2.17 (1.71–2.71)	2.99 (2.45–3.65)	1.91 (1.44–2.49)	2.46 (1.74–3.15)	2.55 (2.03–3.07)	2.82 (2.17–3.53)	2.74 (2.35–3.78)	2.74 (2.35–3.41)	2.67 (2.40–3.51)	NA
**FVC (%ref)**	72 (58–87)	75 (62–86)	63 (49–75)	85 (71–102)	60 (45–70)	70 (56–81)	73 (57–88)	82 (67–96)	82 (66–96)	82 (66–91)	80 (66–108)	NA
**TLCO (mmol/kPa/min)**	3.92 (2.92–5.09)	4.12 (3.08–5.22)	3.22 (2.45–4.71)	4.28 (3.14–5.50)	3.35 (2.36–4.51)	4.44 (3.55–6.06)	3.09 (2.07–4.65)	4.54 (3.05–5.06)	3.83 (2.51–4.50)	4.08 (3.00–5.14)	2.84 (2.62–4.11)	NA
**TLCO (%ref)**	47 (35–59)	47 (36–59)	40 (30–58)	50 (37–63)	42 (32–52)	55 (44–73)	35 (24–50)	50 (36–60)	43 (32–55)	50 (35–59)	40 (29–48)	NA

Data are presented as n (%) or median (interquartile range). NA: not applicable. FVC, forced vital capacity; LuTX, lung transplantation; TLCO, transfer factor of the lung for carbon monoxide.

## Data Availability

The data presented in this study are available upon request from the corresponding author.

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
