# Peer review of "Lung Transplantation in Idiopathic Pulmonary Fibrosis Patients in the European MultiPartner IPF Registry: Challenges for Health Equity"

_biomedicines, 2025, doi:10.3390/biomedicines13112684_

Round 1

Reviewer 1 Report

Comments and Suggestions for Authors

This study fills the gap in real-world data on lung transplantation for patients with idiopathic pulmonary fibrosis (IPF) in Central and Eastern European(CEE). The differences in transplantation rates among various countries revealed are shocking, providing strong evidence for improving the fairness of treatment for patients with end-stage IPF in this region. Secondly, this study is based on the European MultiPartner IPF Registry (EMPIRE), which is a high-quality multinational prospective registry with a large sample size (n=4,390). This large sample size enhances the statistical power and representativeness of the research. Meanwhile, the study combined objective registry data analysis with subjective expert questionnaire surveys to explore the factors influencing referral and transplantation from both patient characteristics and clinical practice levels, making the analysis more in-depth and comprehensive. However, further clarification is needed on the following issues to enhance the reliability and explanatory power of the results.

  1. The title and abstract do not explicitly state that this is a retrospective cohort study. Readers are required to make their own judgments based on some parts of the abstract. It is suggested that improvements be made so that readers can more quickly grasp the design nature of the study.
  2. In the "Materials and Methods" section, a brief description of the research type will help readers clearly understand the overall research framework from the very beginning.
  3. The IPF patient data collected in the article were uploaded to EMPIRE by each center separately. There is a lack of a brief description of the consistency of data measurement methods and standards among the centers. It is suggested that the authors improve it to further enhance the comparability of the data and the credibility of the results.
  4. In this study, there were varying degrees of data deficiency in key pulmonary function and exercise endurance indicators (such as forced vital capacity (FVC), carbon monoxide diffusion capacity (TLCO), 6-minute walking distance, etc.), which might affect the accuracy of the analysis of differences between groups. If the measures taken to address this issue could be briefly described in the methodology section, it would enhance the rigor of the research methods.
  5. There are certain limitations in the response rate of the questionnaire used in this study (for example, only 5 countries participated in the ranking of referral factors, and only 4 countries participated in the free response part). The differences among the countries that did not respond may introduce selection bias. For instance, the participation of Israel, a country with a high transplantation rate, might lead the outcome to lean towards a "successful experience" perspective.
  6. In the results section (Section 3.2 and Table 3), the unadjusted comparison results of baseline characteristics between the transplant group and the non-transplant group are presented. However, there may be complex interrelationships among various factors, and current univariate analyses cannot clarify which factors are independent predictors for patients' eventual acceptance of transplantation. To enhance the scientific rigor of the research and make the conclusions more instructive for clinical practice and decision-making, it is suggested that the authors supplement multivariate statistical analysis.

Author Response

Dear Reviewer 1,

please find attached the revised version of our manuscript entitled „Lung Transplantation in Idiopathic Pulmonary Fibrosis Patients in the European MultiPartner IPF Registry: Challenges for Health Equity”.

We are grateful for the effort of the Reviewers and the Editorial Office, and we hope you find our responses satisfactory.

Please find our response to your comment below.

Yours sincerely, Authors

Comment 1: The title and abstract do not explicitly state that this is a retrospective cohort study. Readers are required to make their own judgments based on some parts of the abstract. It is suggested that improvements be made so that readers can more quickly grasp the design nature of the study.

Response 1:  We fully agree with the Reviewer that the retrospective nature of our study is not clearly indicated in the title and abstract. To address this, we modified the abstract to highlight the retrospective study design, as follows: „Baseline characteristics of IPF patients potentially eligible for LuTX, enrolled into the European MultiPartner IPF Registry between 2012 and 2022 (n = 1,256) were retrospectively analyzed” (page 1, line 30).

Comment 2: In the "Materials and Methods" section, a brief description of the research type will help readers clearly understand the overall research framework from the very beginning.

Response 2: We agree with the Reviewer that the retrospective design is not sufficiently emphasized in the “Materials and Methods” section, which could hinder readers’ understanding of our research procedure. Therefore, we added the following to the “Study Design” subsection of “Materials and Methods” and moved this subsection to the beginning of the section for better readability: “In our retrospective cohort analysis LuTX referral and listing practices of the EMPIRE countries were evaluated by applying the ISHLT selection criteria for LuTX candidates [4] (adapted to the format and availability of data in the registry) to the baseline data from all patients enrolled into the registry.” (page 2, line 77).

Comment 3: The IPF patient data collected in the article were uploaded to EMPIRE by each center separately. There is a lack of a brief description of the consistency of data measurement methods and standards among the centers. It is suggested that the authors improve it to further enhance the comparability of the data and the credibility of the results.

Response 3: We appreciate the Reviewer’s comment. The patients’ data included in the EMPIRE registry were obtained during routine clinical assessments at each participating center. Although data entry was performed locally, all centers are expected to follow internationally accepted guidelines and standardized procedures for the diagnosis and management of IPF. Therefore, the collected data are considered comparable across centers.

Comment 4: In this study, there were varying degrees of data deficiency in key pulmonary function and exercise endurance indicators (such as forced vital capacity (FVC), carbon monoxide diffusion capacity (TLCO), 6-minute walking distance, etc.), which might affect the accuracy of the analysis of differences between groups. If the measures taken to address this issue could be briefly described in the methodology section, it would enhance the rigor of the research methods.

Response 4: We thank the Reviewer for this comment. No specific correction methods were applied to address missing data. However, the proportion of missing data was similar across the different study groups, suggesting that the potential impact on the comparative analyses between groups is likely minimal.

Comment 5: There are certain limitations in the response rate of the questionnaire used in this study (for example, only 5 countries participated in the ranking of referral factors, and only 4 countries participated in the free response part). The differences among the countries that did not respond may introduce selection bias. For instance, the participation of Israel, a country with a high transplantation rate, might lead the outcome to lean towards a "successful experience" perspective.

Response 5: The Reviewer raised an important point that the relatively low response rate of the questionnaire limits the generalizability of our findings across all EMPIRE countries. However, the countries with the largest numbers of transplanted patients in the registry did respond, indicating that age and baseline comorbidities are still major exclusion factors in the leading countries of the region. It should be noted that in non-responder countries with very low or no LuTX activity, additional factors limiting accessibility may exist, which are not captured in our study. To address this, we added the following sentence to the limitations in the Discussion section: “The questionnaire analysis provides only a limited representation of the different healthcare systems within EMPIRE due to a lack of responses from a substantial number of the participating countries.” (page 12, lines 281-284).

Comment 6: In the results section (Section 3.2 and Table 3), the unadjusted comparison results of baseline characteristics between the transplant group and the non-transplant group are presented. However, there may be complex interrelationships among various factors, and current univariate analyses cannot clarify which factors are independent predictors for patients' eventual acceptance of transplantation. To enhance the scientific rigor of the research and make the conclusions more instructive for clinical practice and decision-making, it is suggested that the authors supplement multivariate statistical analysis.

Response 6: We appreciate the Reviewer’s attention to this methodological aspect. The comparison of baseline characteristics is presented as more of an exploratory analysis. A multivariate model, such as binomial regression, would not be sufficiently powered to draw clinically relevant conclusions, particularly because of the limited number of transplanted patients and because the eligibility criteria for transplantation in real life include numerous and interrelated variables. Interactions between age and lung function, comorbidities, medications or GAP index; interactions between disease severity and therapy modalities; smoking and lung function or comorbidities are well recognized and they were not in the focus of our study. We believe that the univariate comparisons provide valuable context for clinical interpretation regarding the key trends relevant to transplant eligibility. Finally, we think that an overcomplicated baseline comparison would draw attention away from the main clinical message of the article.

Reviewer 2 Report

Comments and Suggestions for Authors

In terms of retrogressively data analysis on Lung transplantation and its outcomes with different perspectives and their challenges are encouraging. However, my few concerns are as below:

1. In Figure 2, the author may use contrasting colors to show the bar diagram. 

2. The author may add data where data reflects the cause of the morbidity of LuTX patients and their average duration after LuTX, which will demonstrate the exclusion and inclusion relevance with the LuTX and health outcome.

3. Data is not conclusive to show the challenges related to Lung Transplantation in Idiopathic Pulmonary Fibrosis 2
Patients. The author may use data from the United States to give baseline statistics.

4. In Figure 2, the author shows a large number of eligible patients for LuTX not transplanted. However, in the discussion section, the author didn't justify their findings. 

Author Response

Dear Reviewer 2,

please find attached the revised version of our manuscript entitled „Lung Transplantation in Idiopathic Pulmonary Fibrosis Patients in the European MultiPartner IPF Registry: Challenges for Health Equity”.

We are grateful for the effort of the Reviewers and the Editorial Office, and we hope you find our responses satisfactory.

Please find our response to your comment below.

Yours sincerely, Authors

Comment 1: In Figure 2, the author may use contrasting colors to show the bar diagram. 

Response 1: We thank the Reviewer for this helpful suggestion. To improve the visual clarity of Figure 2, we have adjusted the color scheme of the bar diagram accordingly.

Comment 2: The author may add data where data reflects the cause of the morbidity of LuTX patients and their average duration after LuTX, which will demonstrate the exclusion and inclusion relevance with the LuTX and health outcome.

Response 2:  We thank the Reviewer for this valuable comment. The EMPIRE registry is dedicated to patients with idiopathic pulmonary fibrosis; therefore, the underlying cause of morbidity was uniform across the cohort. LuTX was defined as an endpoint for follow-up in the registry, meaning that no post-transplant data (including outcomes or duration after LuTX) are available. This is described among the limitations in the Discussion section as follows: “Another limitation was that LuTX was a reason to stop follow-ups of patients in the registry, i.e. only data up to the date of LuTX are available for the transplanted cohort.”

Comment 3: Data is not conclusive to show the challenges related to Lung Transplantation in Idiopathic Pulmonary Fibrosis 2
Patients. The author may use data from the United States to give baseline statistics.

Response 3:  We thank the Reviewer for this comment. The present study was based on data from the EMPIRE registry, which specifically includes idiopathic pulmonary fibrosis (IPF) patients from Central and Eastern European countries where data on lung transplantation in IPF have previously been limited or unavailable. Therefore, U.S. data were not included, as the aim of our analysis was to characterize factors influencing LuTX referral and eligibility within the EMPIRE cohort rather than to compare outcomes across different international datasets.

Comment 4: In Figure 2, the author shows a large number of eligible patients for LuTX not transplanted. However, in the discussion section, the author didn't justify their findings. 

Response 4: We acknowledge the Reviewer’s concern and would like to clarify this point. The primary focus of our study was to explore why a considerable proportion of patients potentially eligible for LuTX were not transplanted, and the Discussion section elaborates on the possible explanations for this observation in detail. As stated in the Discussion:

“In line with previous data, advanced age and comorbidities were shown as the main reasons for not referring patients in the EMPIRE countries. Although age higher than 70 years is considered as a factor with high risk for poor LuTX outcome, there is no age limit as an absolute contraindication according to current international guidelines. […] Life expectancy in CEE countries is variable and generally lower than in more developed countries [15]. As the post-transplant median survival in IPF is estimated around 4.5 years [16], most of these countries use lower age limits as a relative contraindication for transplant. However, this conservative age approach in CEE countries contradicts emerging evidence demonstrating meaningful survival benefit for single LuTx in IPF patients over 65 years of age, with no difference in survival following transplant, rates of primary graft dysfunction, or acute/chronic rejection when compared to younger recipients [17]. Our finding that age >70 years remains a leading exclusion criterion suggests CEE countries may be missing opportunities to extend survival for a significant patient population.”

Also “Considering that a substantial proportion (30.9%) of the transplanted patients were initially not considered for LuTX (see Table 2) partly due to comorbidities, partly due to other reasons, yet were eventually transplanted, there could be a subjective overemphasis on comorbidities by the treating physicians. It would be important for referring physicians to more systematically track patients’ inclusion and exclusion criteria and to re-evaluate them over the course of the disease, particularly in light of the adequate management of comorbid conditions.”

Furthermore “Although double LuTX is preferred in most CEE countries, accumulating evidence suggest no significant difference in overall survival between the two techniques [18, 19, 21], particularly among older recipients [17-20, 24]. In Israel, single LuTX is more commonly performed in IPF than double LuTX [23, 25]. This could explain their notable success in achieving high LuTx rates, and may serve as a direction to follow in the CEE region in order to broaden the scope of LuTX candidates in IPF patients, even in the age group over 65 years.”